**Subject Area:**
biochemistry

Tn5 transposase, chromatin, nucleosome, linker DNA, nucleosome-free region

**Author for correspondence:**
Hitoshi Kurumizaka
e-mail: kurumizaka@iam.u-tokyo.ac.jp

†Present address: Laboratory of Chromosome and Cell Biology, The Rockefeller University, 1230 York Ave, New York, NY 10065, USA.

# Biochemical analysis of nucleosome targeting by Tn5 transposase

Shoko Sato[1], Yasuhiro Arimura[1,†], Tomoya Kujirai[1], Akihito Harada[2], Kazumitsu Maehara[2], Jumpei Nogami[2], Yasuyuki Ohkawa[2] and Hitoshi Kurumizaka[1]

[1]Laboratory of Chromatin Structure and Function, Institute for Quantitative Biosciences, The University of Tokyo, 1-1-1 Yayoi, Bunkyo-ku, Tokyo 113-0032, Japan
[2]Division of Transcriptomics, Medical Institute of Bioregulation, Kyushu University, 3-1-1 Maidashi, Higashi-ku, Fukuoka 812-8582, Japan

HK, 0000-0001-7412-3722

Tn5 transposase is a bacterial enzyme that integrates a DNA fragment into genomic DNA, and is used as a tool for detecting nucleosome-free regions of genomic DNA in eukaryotes. However, in chromatin, the DNA targeting by Tn5 transposase has remained unclear. In the present study, we reconstituted well-positioned 601 dinucleosomes, in which two nucleosomes are connected with a linker DNA, and studied the DNA integration sites in the dinucleosomes by Tn5 transposase *in vitro*. We found that Tn5 transposase preferentially targets near the entry–exit DNA regions within the nucleosome. Tn5 transposase minimally cleaved the dinucleosome without a linker DNA, indicating that the linker DNA between two nucleosomes is important for the Tn5 transposase activity. In the presence of a 30 base-pair linker DNA, Tn5 transposase targets the middle of the linker DNA, in addition to the entry–exit sites of the nucleosome. Intriguingly, this Tn5-targeting characteristic is conserved in a dinucleosome substrate with a different DNA sequence from the 601 sequence. Therefore, the Tn5-targeting preference in the nucleosomal templates reported here provides important information for the interpretation of Tn5 transposase-based genomics methods, such as ATAC-seq.

## 1. Introduction

Chromatin is the eukaryotic nuclear architecture by which genomic DNA is highly compacted and accommodated within a nucleus [1]. In chromatin, the core histones H2A, H2B, H3 and H4 form the histone octamer, and approximately 150 base-pairs of DNA are bound to the histone octamer surface. Consequently, the DNA stretch is left-handedly wrapped around the histone octamer, thus forming the nucleosome [1–3]. The nucleosomes are connected with the linker DNA segments in chromatin. In the nucleus, the linker DNA lengths are not uniform, depending on the genomic loci and cell types, and are determined by the translational positions of the nucleosomes [4–10]. The DNA directly bound to the histone surface within the nucleosome is usually inaccessible to DNA-binding proteins, which function as regulators of transcription, replication, repair and recombination [10–13]. The histone-free linker DNA regions then become the target sites for these DNA-binding proteins [14,15]. Therefore, the nucleosome positioning is an important regulatory element for genomic DNA compaction and regulation.

To probe the nucleosome positioning in cells, nuclease hypersensitivity, in which the DNA regions without nucleosomes (nucleosome-free regions, NFRs) and the linker DNA regions between nucleosomes are preferentially digested, is commonly used. Deoxyribonuclease I (DNase I) is a double-stranded endonuclease that is employed for detecting the nucleosome-free DNA regions in the genome [16,17]. Micrococcal nuclease is an endo-exonuclease that preferentially

digests the DNA stretches without nucleosomes, such as linker DNAs, and is used to analyse nucleosome occupancy and positioning in cells [10,16,17]. These nucleosome-mapping methods are combined with high-throughput DNA sequencing, and are commonly used for chromatin analysis, but usually require multiple steps and/or a large number of cells [16,17].

The assay for transposase-accessible chromatin using sequencing (ATAC-seq) method has been developed for detecting NFRs and linker DNA regions by a simple procedure from small amounts of input materials, based on the activity of Tn5 transposase, followed by high-throughput DNA sequencing [18]. The Tn5 transposase technology also enables the mapping of the distributions of histone modifications and DNA-binding proteins in 100–1000 cells by the chromatin integration labelling method [19] and in greater than 60 cells by the CUT&Tag method [20]. In these methods, the genomic DNA sequences associated with the target molecules are tagged with adaptor DNA sequences by the Tn5 transposase-mediated integration.

Bacterial Tn5 transposase promotes the transposition of the Tn5 transposon [21]. The Tn5 transposon contains two inverted repeats, IS50L and IS50R, and each IS50 repeat is located adjacent to two different 19-base-pair Tn5 transposase recognition sequences: the outside end and inside end sequences are located at the Tn5 transposon–host genomic DNA boundary and in the Tn5 transposon, respectively. Tn5 transposase is encoded in IS50R, and catalyses the Tn5 transposition in bacterial cells. The first step of the transposition reaction is the formation of a synaptic complex containing a Tn5 transposase dimer with two outside end sequences of the Tn5 transposon. Tn5 transposase excises the transposon from the flanking genomic DNA, via hairpin formation with the DNA ends of the transposon, and releases the synaptic complex, resulting in a pair of 3′-OH groups at the blunt ends of the excised transposon DNA fragment [22]. The 3′-OH groups of the DNA attack the phosphodiester bonds of the target DNA, integrating the transposon in the target site within the synaptic complex [23,24]. In the integration step, Tn5 transposase cleaves the target DNA with nine nucleotide 5′ overhangs, and the resulting nine base-pair gaps flanking the transposon are filled by the cellular DNA polymerase [25]. *In vitro*, Tn5 transposase effectively catalyses the transposition using synthetic oligonucleotide sequences (adaptor DNAs), under conditions with $Mg^{2+}$ ions [26,27].

The ATAC-seq method has been applied to map the nucleosome positioning and relaxed 'active' chromatin regions, which correlate to the DNase I hypersensitive sites, in diverse cell types and developmental stages [17,18,28–32]. However, it has remained unclear how Tn5 transposase targets the locations of the adaptor DNA integration sites in the chromatin substrates. In the present study, we reconstituted dinucleosomes with various linker DNA lengths, and mapped the Tn5 transposase-mediated adaptor DNA integration sites in model chromatin substrates.

# 2. Results

## 2.1. Tn5 transposase integrates DNA fragments at a specific site in the nucleosomal DNA

To test the targeting sites of Tn5 transposase in chromatin, we performed Tn5 transposase assays with nucleosomal DNA substrates *in vitro*. Purified Tn5 transposase was incubated with short oligonucleotide duplexes (adaptor DNAs), and the Tn5 complexed with adaptor DNAs (Tn5–DNA complex) was purified by gel filtration chromatography (electronic supplementary material, figure S1A–C). The adaptor DNA integration reaction by Tn5 transposase was conducted with the reconstituted dinucleosome as the targeting substrate (figure 1a). The dinucleosome was reconstituted on the 601 sequence, which uniquely forms a nucleosome at a single position (figure 1b). In the dinucleosome, two nucleosomes were connected with a 15-base-pair linker DNA (figure 1b, upper panel). In this experimental system, the reaction products can be detected as DNA fragments, because Tn5 cleaves the target DNA and integrates the adaptor DNAs at the cleaved site (figure 1a). The DNA fragments were detected by non-denaturing polyacrylamide gel electrophoresis (PAGE).

In the presence of the naked DNA substrate, the Tn5–DNA complex generated multiple DNA fragments, which represented the integration of the adaptor DNAs into different sites of the target DNA (figure 1c, lanes 2–6). On the other hand, in the presence of the dinucleosome substrate, the Tn5–DNA complex appeared to cleave a single site, resulting in two DNA fragments (figure 1c). These data suggested that the adaptor DNAs were integrated at a specific site (figure 1c, lanes 8–12). Short minor products were observed, when the reactions were conducted for longer times (figure 1c, lanes 9–12). These minor products may correspond to the products resulting from the Tn5 integration periodicity in chromatin, as previously reported [31]. A massively parallel sequencing analysis confirmed the detailed cleavage sites. The major cleavage sites are between positions 140 and 141 from one end of the dinucleosomal DNA, together with the site between positions 149 and 150 (Tn5 cleaves the target DNA with nine nucleotide 5′ overhangs) (figure 1d,e, middle panel). Therefore, the major cleavage site was mapped at the 5′–141 base and 5′–149 base positions, near the entry–exit sites of the proximal nucleosomal DNA (figure 1e, middle panel). Additionally, the minor cleavage sites are also mapped at the 5′–140 base and 5′–148 base positions and at the 5′–142 base and 5′–150 base positions (figure 1d,e, top and bottom panels).

## 2.2. Effects of linker length on the integration reaction by Tn5 transposase

We next tested the effects of the linker length on the Tn5 integration reaction. The dinucleosomes with various linker DNA lengths (0, 5, 10, 15, 20, 25 and 30 base-pairs) were prepared based on the 601 sequence (figure 2a; electronic supplementary material, figure S2A and B), and then the integration reactions were performed with the Tn5–DNA complex. As shown in figure 2b, in the presence of the dinucleosome with the 0 base-pair linker DNA, the resulting DNA fragments were hardly detected, suggesting that the Tn5–DNA complex did not efficiently promote the integration reaction in chromatin without a linker DNA (lanes 2,3). On the other hand, in the presence of the dinucleosome containing the 5-base-pair linker DNA, the Tn5–DNA complex cleaved the dinucleosome substrate (figure 2b, lanes 4,5). These data indicated that Tn5 transposase requires a linker DNA region for the integration reaction into the nucleosomal DNA. Surprisingly, the integration reaction by Tn5 transposase was drastically enhanced when the linker DNA length was expanded to 15 base-pairs (figure 2b, lanes 8,9). The enhancement was not obvious with the dinucleosome

royalsocietypublishing.org/journal/rsob   Open Biol. 9: 190116

royalsocietypublishing.org/journal/rsob Open Biol. 9: 190116

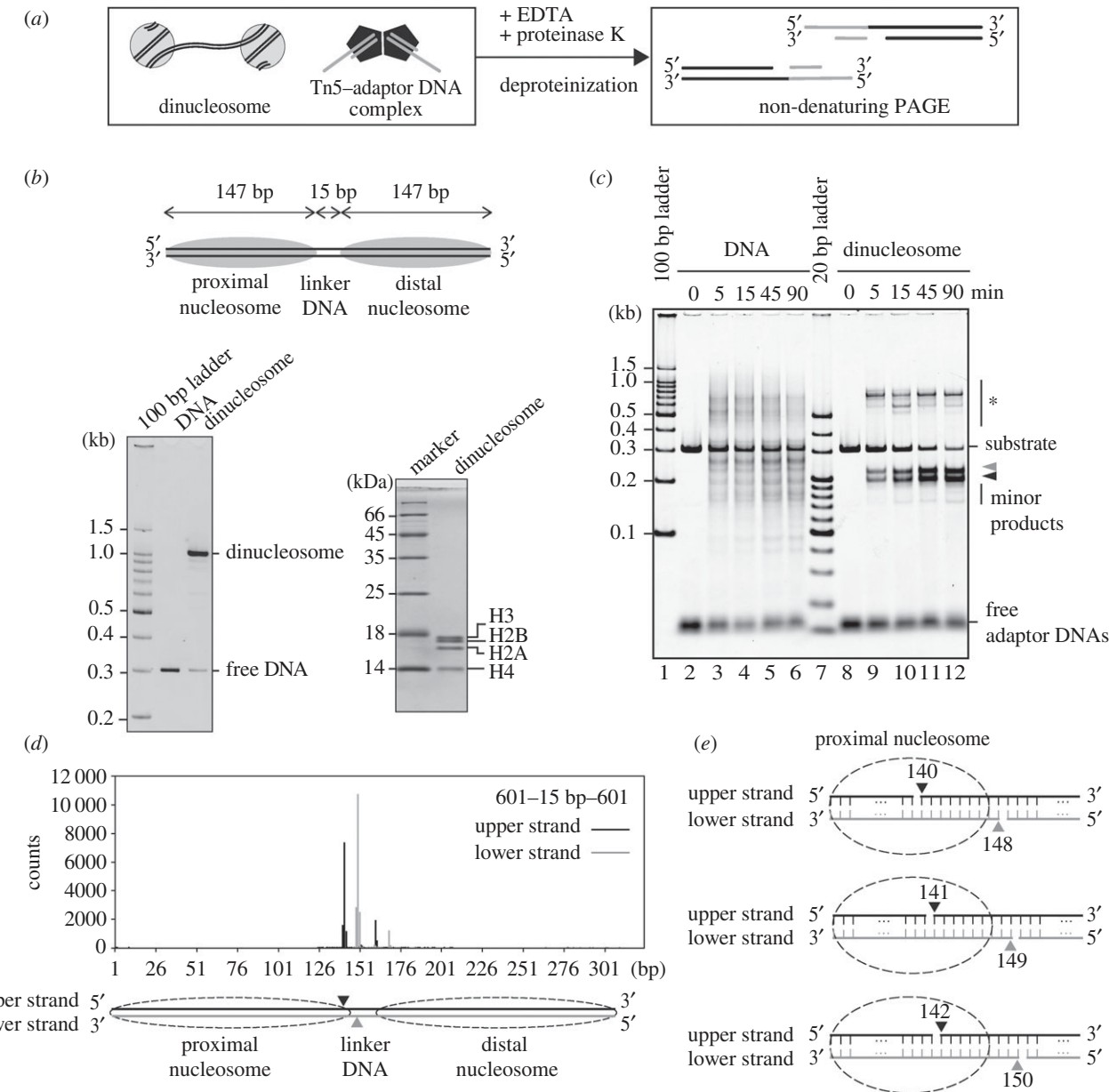

**Figure 1.** Tn5 integrates DNA fragments at specific sites of the nucleosomal DNA. (*a*) Schematic of a Tn5 transposase assay with dinucleosome substrates. (*b*) Schematic of the dinucleosome containing the 15-base-pair linker DNA (upper panel). The reconstituted dinucleosome was analysed by non-denaturing PAGE with ethidium bromide staining (left panel) and denaturing PAGE with Coomassie Brilliant Blue staining (right panel). (*c*) Time course of the adaptor DNA integration reaction by Tn5 transposase. The naked DNA or the dinucleosome containing the 15-base-pair linker DNA (containing 0.01 µg µl$^{-1}$ DNA) was incubated with the Tn5–DNA complex (0.5 µM). After deproteinization, DNA fragments were analysed by non-denaturing PAGE with SYBR Gold staining. The bands marked with * are annealing products of the DNA fragments containing partially single-stranded DNA. Grey and black arrowheads indicate longer and shorter DNA fragments produced by the Tn5 transposase reaction, respectively. (*d*) A profile of the Tn5 transposase cleavage sites for the dinucleosome containing the 15-base-pair linker DNA. The DNA fragments tagged by the Tn5 transposase assay were analysed by massively parallel paired-end sequencing, and the 5′-ends of the DNA fragments are mapped on the substrate DNA sequence, as described in the Methods. Major cleavage sites are shown by arrowheads in the schematic. (*e*) Schematic of the major cleavage sites of the dinucleosome containing the 15-base-pair linker DNA.

containing the 10-base-pair linker DNA (figure 2*b*, lanes 6,7). The Tn5–DNA complex exhibited efficient cleavage activity with the dinucleosomes containing the 20-, 25- and 30 base-pair linker DNAs (figure 2*b*, lanes 10–15). Therefore, a certain linker DNA length (10–15 base-pairs) is an important factor for the efficient Tn5 integration reaction in chromatin.

## 2.3. Tn5 targets a specific nucleosomal DNA site independent of the linker DNA length

As shown in figure 2*b*, the lengths of the resulting longer DNA fragments were increased in a linker length-dependent manner

(black arrowhead). By contrast, the lengths of the resulting shorter DNA fragments were the same among the integration reactions with the dinucleosomes containing 5-, 10-, 15-, 20-, 25- and 30-base-pair linker DNAs (figure 2*b*, grey arrowhead). These results indicated that the Tn5–DNA complex cleaves a specific nucleosomal DNA site around the entry–exit sites of the nucleosome, in the dinucleosome substrates containing different linker DNA lengths (figure 2*b*). When the Tn5–DNA complex reacted with the dinucleosome containing the 30-base-pair linker DNA, an additional band was observed (figure 2*b*, white arrowhead). A massively parallel sequencing analysis revealed that this additional cleavage site was mapped at the 5′–160 base and 5′–168 base positions, which

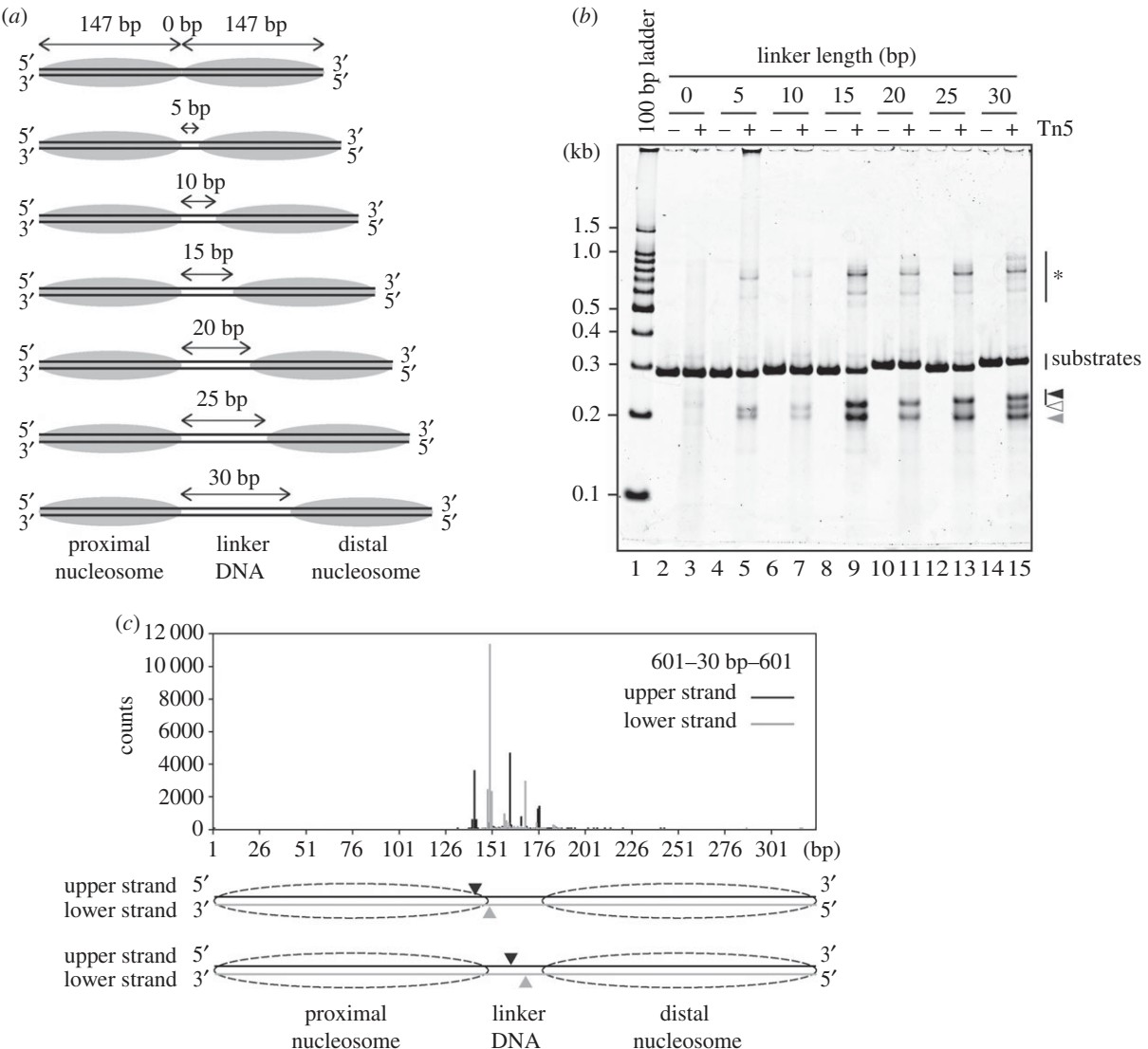

**Figure 2.** Effects of linker length on the integration reaction by Tn5 transposase. (*a*) Schematic of the dinucleosome substrates. (*b*) The dinucleosomes containing the 0-, 5-, 10-, 15-, 20-, 25- or 30-base-pair linker DNAs (containing 0.01 µg µl⁻¹ DNA) were incubated with the Tn5–DNA complex (0.5 µM). After deproteinization, the DNA fragments were analysed by non-denaturing PAGE with SYBR Gold staining. The bands marked with * are annealing products of the DNA fragments containing partially single-stranded DNA. Black, white and grey arrowheads indicate longer, middle and shorter DNA fragments produced by the Tn5 transposase reaction, respectively. The substrate DNAs exhibited unusual migration profiles, which do not correspond to the DNA length, probably due to the structural nature of the 601 sequence. The DNA sequences of these template DNAs were confirmed by direct sequencing. (*c*) Profile of the Tn5 transposase cleavage sites for the dinucleosome containing the 30-base-pair linker DNA. The DNA fragments tagged by the Tn5 transposase assay were analysed by massively parallel paired-end sequencing, and the 5'-ends of the fragments are mapped on the substrate DNA sequence, as described in the Methods. The two major cleavage sites are shown in the schematic.

are located in the middle of the linker DNA region (figure 2*c*; electronic supplementary material, table S1). This additional cleavage site was not observed as a major site for the naked DNA template (electronic supplementary material, table S1), suggesting that the dinucleosome formation with the 30-base-pair linker DNA may dictate the cleavage site in the linker DNA. These findings indicate that the middle of the linker DNA region, which may not directly contact the histone surface, may be additionally cleaved by Tn5 transposase, when the linker DNA length is expanded to around 30 base-pairs.

## 2.4. Effect of the dinucleosomal DNA sequence on the integration reaction by Tn5 transposase

To test whether the DNA sequence affects the integration site in the dinucleosome, the integration reaction by the Tn5–DNA

complex was conducted using dinucleosome substrates containing the 603 sequence, which is different from the 601 sequence (electronic supplementary material, figure S3A). Two 603 dinucleosome substrates with 15-base-pair and 30-base-pair linker DNAs were prepared (figure 3*a*; electronic supplementary material, figure S3B and C). In the Tn5 transposase integration assay with naked DNAs, several discrete bands were detected in the naked 603 DNA substrates, and the band patterns were different from those of the naked 601 DNA substrates (figure 3*b*, lanes 3,7,11,15; electronic supplementary material, table S1). This indicated that the Tn5–DNA complex may exhibit a DNA sequence preference in the integration reaction, consistent with previous reports [18,33,34]. However, in both the 601 and 603 DNA experiments, the specific bands observed in the integration reactions with the dinucleosome substrates were different from those of the naked DNA substrates (figure 3*b*). Therefore, the sequence preference of the

royalsocietypublishing.org/journal/rsob Open Biol. **9**: 190116

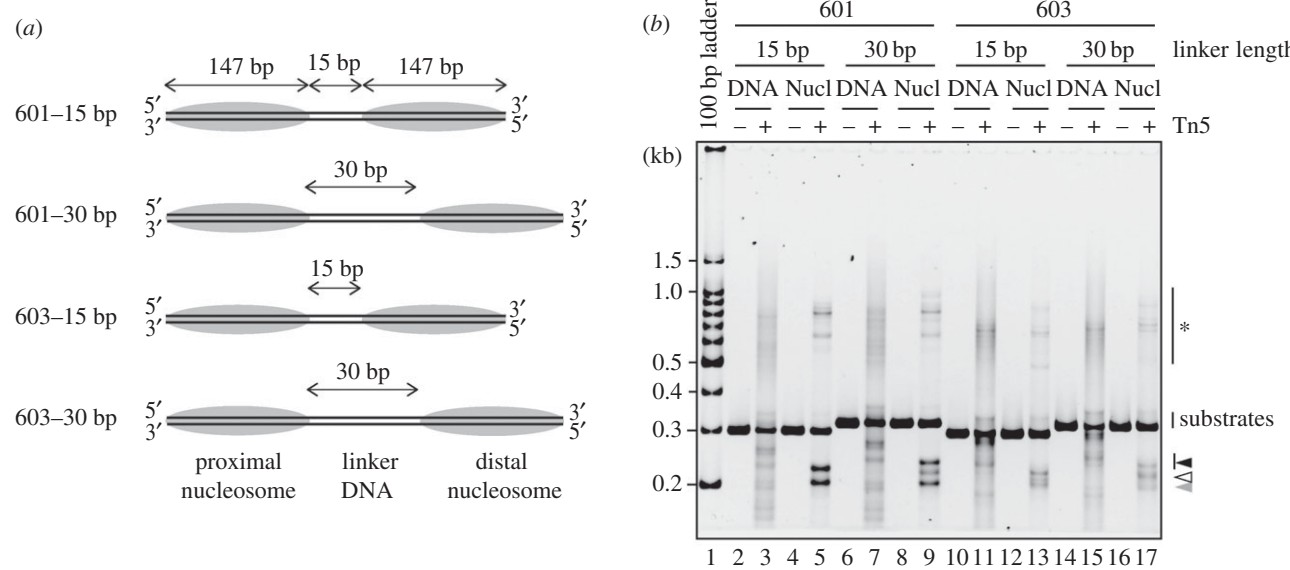

**Figure 3.** Effect of the DNA sequence on the integration reaction by Tn5 transposase. (*a*) Schematic of the dinucleosome substrates containing the 601 or 603 sequences. (*b*) The naked DNA (lanes 3, 7, 11, 15) or dinucleosomes (containing 0.01 µg µl$^{-1}$ DNA, lanes 5,9,13,17) were incubated with the Tn5–DNA complex (0.5 µM). After deproteinization, the DNA fragments were analysed by non-denaturing PAGE with SYBR Gold staining. The bands marked with * are annealing products of the DNA fragments containing partially single-stranded DNA. Black, white and grey arrowheads indicate longer, middle and shorter DNA fragments produced by the Tn5 transposase reaction, respectively.

Tn5–DNA complex may not be a determinant for the specific integration site in the dinucleosome. Intriguingly, the band patterns of the 601 dinucleosomes (15-base-pair and 30-base-pair linker DNAs) were quite similar to those of the 603 dinucleosomes (figure 3*b*, lanes 5,13 and 9,17). These results suggested that, in the dinucleosome substrates, the DNA targeting sites of the Tn5–DNA complex may be dictated by the local DNA situation induced by the nucleosome formation, but not the DNA sequence preference.

It should be noted that, in the 601 dinucleosome substrate, the Tn5 transposase targeting site is restricted to the proximal nucleosome (figures 1*d* and 2*c*). However, in the 603 dinucleosome, the proximal and distal nucleosomes were equally targeted by Tn5 transposase (figure 4*a,b*). The region of the 601 sequence containing the preferential Tn5 transposase targeting site reportedly has lower affinity than the other part [35,36]. Therefore, these results suggested that Tn5 transposase may preferentially target the DNA region flexibly detached from the histone surface in the nucleosome.

## 3. Discussion

In the ATAC-seq method, Tn5 transposase has been employed to insert an adaptor DNA fragment in the nucleosome-free DNA regions and linker DNA regions between nucleosomes. The ATAC-seq signals are obtained as the integration sites of the adaptor DNA, and provide the nucleosome locations in chromatin with small amounts of input materials [18,31,37]. Therefore, the ATAC-seq method has been widely used for chromatin analysis. However, the specific location of the Tn5 transposase targeting site in nucleosomal DNA has not been clarified yet.

In the present study, we performed an *in vitro* Tn5 transposase assay with reconstituted dinucleosomes, and found that Tn5 transposase preferentially integrates adaptor DNAs near the entry–exit sites of the nucleosomal DNA. One DNA site cleaved by Tn5 transposase is located within the histone–

DNA contact region of the nucleosome (figures 1*d*, 2*c* and 4). This Tn5 transposase target site did not depend on the linker DNA length (figure 2*b*). It should be noted that the 5′–140 site, which is also found as the major cleavage site in the dinucleosome substrates with the 601 sequence, has been observed as a preferential cleavage site for Tn5 transposase in the 601 naked DNA substrates (electronic supplementary material, table S1). This sequence preference of Tn5 transposase was not observed when the 603 sequence was used as the dinucleosome and naked DNA substrates (figures 3 and 4; electronic supplementary material, table S1). Therefore, the sequence preference may not be a major cause of the specific nucleosome targeting by Tn5 transposase. Why does Tn5 transposase prefer to target the entry–exit sites of the nucleosomal DNA? One plausible explanation is 'nucleosomal DNA breathing' [38]. In the nucleosome, the entry–exit DNA regions spontaneously detach and re-attach on the histone surface [38]. Tn5 transposase may cleave the DNA stretch in the nucleosomal DNA region, when the target site is stochastically detached from the histone surface by the nucleosomal DNA breathing, rendering this entry–exit DNA region accessible to the enzyme.

In the 601 dinucleosome, the integration reaction by Tn5 transposase occurred in the proximal nucleosome, but rarely occurred in the distal nucleosome (figures 1*d* and 2*c*). This result is consistent with the report that the 601 sequence causes asymmetric nucleosomal DNA breathing, because of the different DNA flexibilities on either side in the DNA sequence [35,36,38]. Interestingly, Tn5 transposase attacks both the proximal and distal nucleosomes, when the 603 sequence is employed (figure 4). The nucleosome containing the 603 sequence may provide attack sites for Tn5 transposase on both sides of the nucleosomal DNA ends, because they may equally bind to the histones.

As discussed above, Tn5 transposase preferentially targeted the entry–exit sites of the nucleosomal DNA, when the linker DNA lengths ranged from 5 to 25 base-pairs (figure 2*b*). However, we also found that, in the presence of the dinucleosome

royalsocietypublishing.org/journal/rsob    Open Biol. **9**: 190116

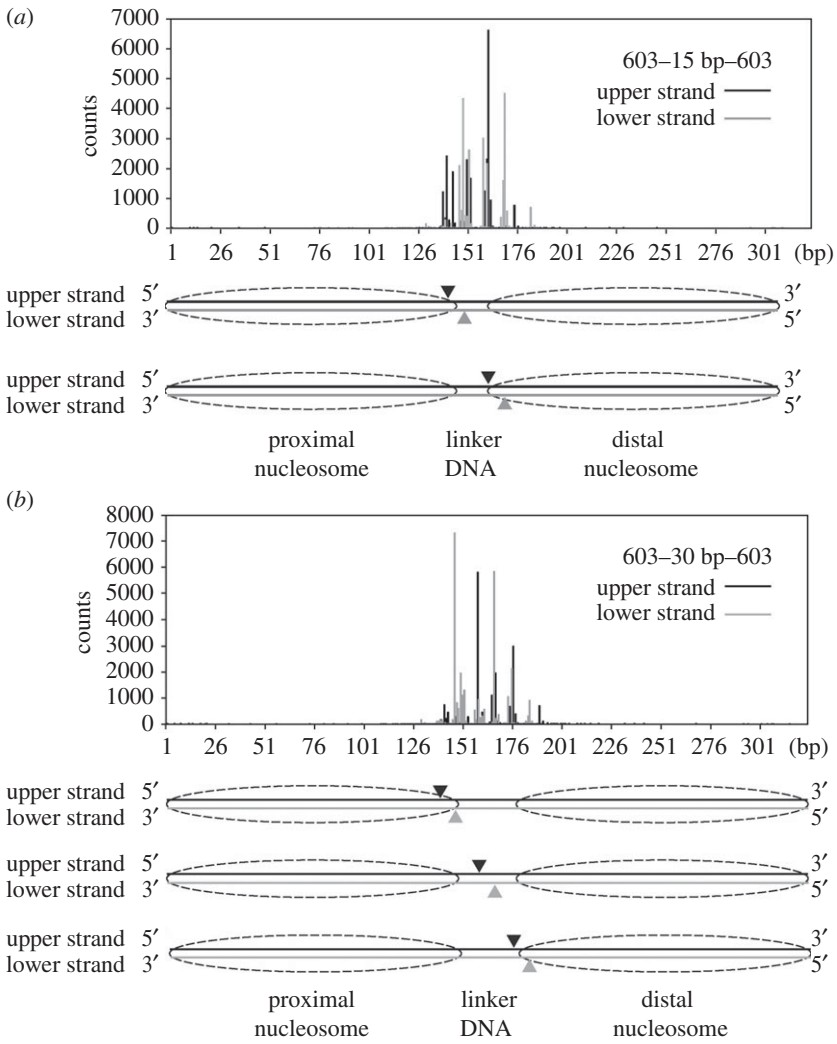

**Figure 4.** The integration sites of the dinucleosome substrates with the 603 sequence. The DNA fragments tagged by the Tn5 transposase assay were analysed by massively parallel paired-end sequencing, and the 5′-ends of the fragments are mapped on the substrate DNA sequence, as described in the Methods. A profile of the Tn5 transposase cleavage sites for the dinucleosomes containing the 15-base-pair linker DNA (a) and the 30-base-pair linker DNA (b). Major cleavage sites are shown by arrowheads in the schematic (a,b).

containing the 30-base-pair linker DNA, Tn5 transposase additionally targeted the middle of the linker DNA (figures 2b and 3b). Therefore, ATAC-seq may specifically detect the nucleosome entry–exit sites, if the linker length is shorter than 30 base-pairs. In cells, the linker DNA length varies among species and genomic regions. Short 5-base-pair linker DNAs exist in mouse embryonic stem cells [9] and yeasts *Saccharomyces cerevisiae* [7] and *Schizosaccharomyces pombe* [8]. In *S. cerevisiae*, the median linker DNA length is reportedly 23 base-pairs [7], and the average linker DNA length around the transcriptional start sites of genes is 18 base-pairs [5]. Schep *et al.* [31] reported the nucleosome positioning of the *S. cerevisiae* genome, using an ATAC-seq-based analysis, and found that the most abundant size of the DNA fragments for nucleosome mapping by Tn5 transposase was 143 base-pairs, which is shorter than the DNA length associated within the nucleosome (145–147 base-pairs). This is consistent with our results, in which Tn5 transposase cleaves at the entry–exit regions within the nucleosomal DNA.

In human primary CD4+ T cells, the average linker DNA lengths have been estimated as approximately 30 base-pairs in the regions retaining the epigenetic marks of active promoters and enhancers, and 58 base-pairs in the regions retaining the heterochromatin marks [6]. In mouse embryonic stem cells, the peaks of the linker DNA length frequencies are reportedly 35 base-pairs and 45 base-pairs [6]. In these cases, Tn5

transposase may attack both the nucleosomal entry–exit and linker DNA regions. These new findings provide important information to decode the ATAC-seq results in cells and/or genomic loci with different linker lengths.

A prototype foamy virus integrase reportedly integrates efficiently into the viral DNA in nucleosomal DNAs [39]. A cryo-electron microscopy structure of the integrase complexed with a nucleosome revealed that the integrase specifically targets the nucleosomal DNA at a position located 3.5 helical turns away from the nucleosomal dyad [39]. In contrast to the foamy virus integrase, Tn5 transposase requires a linker DNA between two nucleosomes for the adaptor DNA integration, and preferentially targets the nucleosomal DNA near the entry–exit site. To clarify the mechanisms by which Tn5 transposase targets the nucleosomal DNA, structural studies of the nucleosome complexed with Tn5 transposase are awaited.

# 4. Methods

## 4.1. Preparation of dinucleosomes

Human histones were prepared as described previously [40]. Briefly, H2A, H2B, H3 and H4, each cloned into the pET15b vector, were expressed as His$_6$-tagged proteins in *Escherichia*

royalsocietypublishing.org/journal/rsob    Open Biol. 9: 190116

*coli*, and purified using Ni-NTA agarose column chromatography (Qiagen) followed by thrombin (Wako) treatment and Mono S column chromatography (GE Healthcare). The histone octamer was reconstituted with purified H2A, H2B, H3 and H4, and was purified by size-exclusion chromatography (Superdex 200 16/60, GE Healthcare), as described previously [40]. The DNA fragments containing the 601 or 603 sequences [41] were cloned into the pGEM-T easy vector (Promega), and the plasmids were prepared from *E. coli* cells. The fragments were excised with an EcoRV treatment, and were prepared by polyethylene glycol precipitation and anion-exchange column chromatography. The DNA fragments were mixed with the histone octamer in 10 mM Tris–HCl (pH 7.5) buffer containing 2 M KCl, 1 mM EDTA and 1 mM dithiothreitol, and the dinucleosomes were reconstituted by continuously decreasing the KCl concentration to 250 mM by the salt-dialysis method [42]. The sequences of these DNA fragments are described in electronic supplementary material, figure S4. The resulting dinucleosomes were fractionated by PAGE using a Prep Cell apparatus (Bio-Rad), as described previously [42]. The dinucleosome concentrations were estimated from the absorbance at 260 nm.

## 4.2. Purification of Tn5 transposase

Tn5 transposase was purified according to the published method, with slight modifications [43]. The plasmid pET21a-Tn5-CBD, which produces the full-length hyperactive Tn5 transposase (E54 K, L372P) as an intein–CBD fusion protein [33,43], was introduced into *E. coli* Rosetta 2 (DE3) cells, and the protein was produced in the presence of 0.25 mM isopropyl β-D-1-thiogalactopyranoside overnight at 10°C, and for an additional 4 h at 23°C. Cells (approx. 10 g) were suspended in 50 ml of 20 mM HEPES-KOH buffer (pH 7.2) containing 0.8 M NaCl, 1 mM EDTA, 10% glycerol, 0.2% Triton X-100 and protease inhibitor cocktail (Nacalai Tesque), and were disrupted by sonication. The cell lysate was centrifuged at 15 000 r.p.m. for 20 min at 4°C. The supernatant was mixed with 5 ml of 10% polyethylenimine (Nacalai Tesque), and the precipitate was removed by centrifugation at 12 000 r.p.m. for 10 min. The supernatant was diluted with 83 ml of the 20 mM HEPES-KOH buffer (pH 7.2) described above, and the precipitate was removed by centrifugation. The resulting supernatant was loaded onto a 17 ml chitin resin (NEB) column, and was washed with the same buffer (20 column volumes), followed by an incubation in 20 mM HEPES-KOH buffer (pH 7.2) containing 100 mM dithiothreitol, 0.8 M NaCl, 1 mM EDTA, 10% glycerol, 0.2% Triton X-100 and protease inhibitor cocktail (Nacalai Tesque) for 36 h. Tn5 transposase was then eluted with the same buffer. The fraction containing Tn5 transposase was dialysed against 100 mM HEPES-KOH buffer (pH 7.2) containing 0.2 M NaCl, 0.2 mM EDTA, 0.2% Triton X-100, 20% glycerol and 2 mM 2-mercaptoethanol, and was loaded on a Mono S 5/50 GL (GE Healthcare) column. After washing the resin with 100 mM HEPES-KOH (pH 7.2) buffer containing 0.2 M NaCl, 0.2 mM EDTA, 20% glycerol and 2 mM 2-mercaptoethanol (five column volumes), Tn5 transposase was eluted by a linear gradient from 0.2 to 1 M NaCl. The purified Tn5 transposase was dialysed against the same buffer (0.2 M NaCl), and was concentrated with a centrifugal concentrator (Millipore). The concentration of Tn5 transposase was determined by ultraviolet measurement with an extinction coefficient at 280 nm (86 525 $M^{-1}$ $cm^{-1}$), and stored at −20°C

in 55 mM HEPES-KOH buffer (pH 7.2) containing 109 mM NaCl, 0.11 mM EDTA, 55% glycerol, 0.85 mM 2-mercaptoethanol and 1.1 mM dithiothreitol (electronic supplementary material, figure S1A).

## 4.3. Preparation of Tn5–adaptor DNA complex

Tn5 transposase complexes with adaptor DNAs containing a hyperactive synthetic sequence (ME sequence, [38]) were purified. Oligonucleotides (Tn5MErev: 5′-[phosphate]-CTGTCT CTTATACACATCT-3′, Tn5ME-A: 5′-TCGTCGGCAGCGTCA GATGTGTATAAGAGACAG-3′, and Tn5ME-B: 5′-GTCTCGT GGGCTCGGAGATGTGTATAAGAGACAG-3′ [43]) were purchased from FASMAC. Tn5MErev and Tn5ME-A, and Tn5MErev and Tn5ME-B were pre-annealed. Tn5 transposase (14 nmol, 1 ml) and oligonucleotide duplexes (Tn5MErev/ Tn5ME-A and Tn5MErev/Tn5ME-B, 42 nmol each, 0.2 ml) were incubated for 1 h at room temperature. The mixture was concentrated to 0.5 ml using a centrifugal concentrator, and was filtered through a 0.22 µm filter. The Tn5 transposase complexed with oligonucleotide duplexes (Tn5–adaptor DNA complex) was separated from the free oligonucleotides by chromatography on a Superdex 200 10/300 GL (GE Healthcare) column, equilibrated with 100 mM HEPES-KOH buffer (pH 7.2) containing 0.2 M NaCl, 0.2 mM EDTA, 20% glycerol and 2 mM dithiothreitol (electronic supplementary material, figure S1B). The concentration of the Tn5–adaptor DNA complex was determined by the Bradford method with Protein Assay CBB Solution (Nacalai Tesque). The Tn5–adaptor DNA complex was stored at a 5 µM concentration in 55 mM HEPES-KOH buffer (pH 7.2) containing 109 mM NaCl, 0.11 mM EDTA, 55% glycerol and 1.1 mM dithiothreitol, at −20°C (electronic supplementary material, figure S1C). For the sequencing analysis, Tn5 transposase complexed with one oligonucleotide duplex (Tn5MErev/Tn5ME-A) was prepared.

## 4.4. Tn5 transposase assay

A naked DNA or a dinucleosome (containing 0.01 µg µl$^{-1}$ for DNA, from 0.05 µg µl$^{-1}$ stock solution) was incubated with the Tn5–adaptor DNA complex (0.5 µM, from 5 µM stock solution described above) in 10 mM *N*-Tris(hydroxymethyl)-methyl-3-aminopropanesulfonic acid-KOH buffer (pH 8.5) containing 1 mM MgCl$_2$ at 37°C. The reactions were stopped by adding stop solution containing 1 mg ml$^{-1}$ Proteinase K (Roche), 0.2% SDS and 20 mM EDTA (final concentration). After an incubation for 30 min at room temperature, the samples were treated with phenol/chloroform/isoamyl alcohol, followed by ethanol precipitation. The DNA fragments were recovered as precipitates, and were resolved in TE (10 mM Tris–HCl, pH 8.0, 0.1 mM EDTA) buffer. The samples were then analysed by non-denaturing 5% PAGE in 0.5× TBE (44.5 mM Tris, 1 mM EDTA, 44.5 mM boric acid) buffer, followed by SYBR Gold staining. The gel images were obtained using an Amersham Typhoon scanner (GE Healthcare; figures 1*c* and 2*b*) or LAS4000 (GE Healthcare; figure 3*b*).

## 4.5. Sequence analysis

The integration reaction was performed with the Tn5–adaptor DNA complex containing Tn5MErev/Tn5ME-A. The DNA or dinucleosome substrate (containing 0.01 µg µl$^{-1}$ DNA, from a 0.05 µg µl$^{-1}$ stock solution) was incubated with the Tn5–

royalsocietypublishing.org/journal/rsob Open Biol. 9: 190116

adaptor DNA complex (601 dinucleosome, 601 naked DNA and 603 dinucleosome: 0.5 µM, from the 5 µM stock solution described above; 603 naked DNA: 0.25 µM, from a 2.5 µM stock solution) in 10 mM $N$-Tris(hydroxymethyl)methyl-3-aminopropanesulfonic acid-KOH buffer (pH 8.5) containing 1 mM $MgCl_2$, at 37°C for 15 min (601 dinucleosome, 601 naked DNA and 603 dinucleosome) or 30 min (603 naked DNA). The resulting DNA fragments were extracted as described above, and they were confirmed by non-denaturing PAGE. The samples were electrophoresed on a TAE-agarose gel, and approximately 100–300 base-pair DNA fragments were purified with a Wizard SV Gel and PCR Clean-Up System (Promega). The purified DNA fragments, in which the adaptor DNA was ligated as a sequence tag by the transposase assay, were analysed by massively parallel sequencing. The DNA fragments (10 ng) were ligated with the annealed Tn5MEDS-B oligonucleotides (Tn5MErev/Tn5ME-B, 2 µM) [43] on the blunt end side at 16°C for 30 min, using a TaKaRa DNA ligation kit. The resulting DNA fragments were purified to exclude fragments shorter than 150 base-pairs, using AMPure XP beads (Beckman Coulter). The polymerase chain reaction (PCR) amplification was performed using the Ad1 (5′-AATGATACGGCGACCACCGAGATCTACACTCGTCGG-CAGCGTCAGATGTG-3′) and Ad2 (5′-CAAGCAGAAGACG GCATACGAGAT[8mer_index]GTCTCGTGGGCTCGGAGAT GT-3′) primers. The PCR reaction was performed under the following conditions: 72°C for 3 min and 95°C for 30 s, followed by seven or eight cycles of 98°C for 10 s, 63°C for 30 s and 72°C for 1 min, with a final extension at 72°C for 5 min, using a Life-ECO thermal cycler (Hangzhou Bioer Technology Co. Ltd., China). The amplified library was purified with a Qiagen MinElute Cleanup kit. The purified library was selected into 260–320 base-pair fragments by E-Gel 2% SizeSelect electrophoresis (Invitrogen). Sequencing was performed in paired reads of 101 × 2 base-pairs, using an Illumina MiSeq system.

## 4.6. Sequencing data analysis

Adaptor DNAs were trimmed using TRIM GALORE (v. 0.5.0) with the following options: –paired –nextera. The paired-end reads were concatenated into single-end reads using FLASH (v. 1.2.11) with the options: -m 15 -M 101 (the reads that have lengths within 101–180 base-pairs were analysed). The reads were mapped to the coordinates of the 601 and 603 DNA sequences using BOWTIE (v. 1.2.2) with the following options: -v0 -m1 (retains exact hits and discards the ambiguous alignments of multi-reads).

Data accessibility. Supplementary figures are available as electronic supplementary material. The deep sequencing data in this study are publicly available at accession no. GEO: GSE130322.

Authors' contributions. S.S., Y.A. and T.K. prepared the materials and performed biochemical analyses, and A.H., K.M., J.N. and Y.O. performed the deep sequencing analysis. H.K. conceived, designed and supervised all of the work, and wrote the paper. All of the authors discussed the results and commented on the manuscript.

Competing interests. The authors declare no competing financial interests.

Funding. This work was funded by a JST CREST grant no. (JPMJCR16G1 to H.K. and Y.O., in part); JSPS KAKENHI (JP17H01408 and JP18H05534 to H.K., in part); the Platform Project for Supporting Drug Discovery and Life Science (Basis for Supporting Innovative Drug Discovery and Life Science Research (BINDS)) from AMED (JP18am0101076 to H.K., in part).

Acknowledgement. We thank Ms. Yukari Iikura (The University of Tokyo) for her assistance, and Mr Satoshi Sekine (Waseda University) for his contribution in the initial stage of this project.

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
