## [Reviewer comments · Open Biology]

Review History

RSOB-19-0116.R0 (Original submission)

Review form: Reviewer 1

Recommendation

Accept with minor revision (please list in comments)

Are each of the following suitable for general readers?

- a) **Title**
Yes
- b) **Summary**
Yes
- c) **Introduction**
Yes

Is the length of the paper justified?

Yes

Should the paper be seen by a specialist statistical reviewer?

No

Is it clear how to make all supporting data available?

Yes

Is the supplementary material necessary; and if so is it adequate and clear?

Yes

Do you have any ethical concerns with this paper?

No

Comments to the Author

This manuscript presents an investigation into Tn5 transposase targeting on defined dinucleosomal templates. It links the high quality nucleosome preparation skills of the Kurumizaka laboratory to create a substrate for analysis of in vitro Tn5 attack on nucleosomes with a next generation sequencing readout.

The methods and results of this study are clear and well-described. The title could be slightly edited to "Biochemical analysis of nucleosome targeting by Tn5 transposase" to better reflect the nature of the work.

However, the depth of interpretation provided is very weak and should make more effort to address the question of dependence of Tn5 site specificity on DNA sequence versus nucleosome structure. This is very important because the extreme bias the authors report of incorporation at only 1-3 sites impacts strongly on broader use and interpretation of ATAC-seq experiments.

Firstly, the relative enhancement/protection by the histone octamer is highly relevant to the interpretation. DNA sequence preferences of Tn5 on free DNA have been published and the authors probably have NGS data for the free DNA substrates they show. Inspection of free DNA bands in figures 1C and 3B suggests that the sites identified as targets with the nucleosome substrate may already be strongly preferred in free DNA. A more detailed analysis could also incorporate data from *S. cerevisiae* mapping by the Greenleaf group.

A related aspect is the spread of incorporation. Figure 1E and the text seems to imply that Tn5 insertion is a single base sites, whereas figure 1D appears shows spread over at least 3 bases on 601-15, and figure 4 shows even more spread on 603. A more detailed description of this is needed.

Secondly, more detailed structural consideration should be included. Since the structure of the 601 containing nucleosome and Tn5 transposase are known, it is reasonable to provide a simple docking schematic to assess the geometry and spatial features observed. The relevance of the co-crystal structure of a nucleosome with prototype foamy virus integrase should be discussed, but has not even been cited. Such a discussion does not need to overshadow the impact of a nucleosome-Tn5 structure that the authors may be attempting to determine.

One smaller issue is over-referencing in blocks of citations: Of the 65 total references, 58 (89%) are cited in the Introduction. On page 2 line 26 there is a block of 13 references for genome-wide nucleosome mapping, on page 3 line 1 there are 4 references for generic genome functionality, on page 3 line 9 there are 5 references for DNaseI mapping, and on page 4 line 15 there are 5 + 12

references for applications of ATAC-seq. In total, these 4 blocks have 39 citations accounting for 60% of all references. With due respect to the cited primary research, many of these references could be subsumed by citing a summary review and a few key primary papers.

There do not seem to be any figure legends for the supplementary figures.

Review form: Reviewer 2

Recommendation

Accept with minor revision (please list in comments)

Are each of the following suitable for general readers?

- a) **Title**
Yes
- b) **Summary**
Yes
- c) **Introduction**

Is the length of the paper justified?

Yes

Should the paper be seen by a specialist statistical reviewer?

No

Is it clear how to make all supporting data available?

Yes

Is the supplementary material necessary; and if so is it adequate and clear?

Not Applicable

Do you have any ethical concerns with this paper?

No

Comments to the Author

Tn5 transposase has been widely used for the mapping of accessible chromatin regions, which are usually the transcription binding sites. However, little attention has been paid on the effect of nucleosome/chromatin structure on the integration of Tn5. In the manuscript titled as "Biochemical analysis of chromatin targeting by Tn5 transposase", the authors explicitly analyzed the effects of in vitro assembled di-nucleosomes with different linker length on the integration of Tn5. They found that Tn5 transposase minimally cleaved the di-nucleosome without a linker DNA and Tn5 transposase preferentially targets near the entry-exit sites of the proximal nucleosomal DNA. In addition to the entry-exit sites of the nucleosome, Tn5 transposase would additionally target the middle of the linker DNA when the linker length of di-nucleosome extended to 30 bp. In conclusion, this study provides useful insights into the mechanisms by which nucleosomes affect Tn5 integration. However, several concerns need to be addressed before this manuscript could be accepted for publication.

Major concerns:

1. The authors observed preferential integration of Tn5 at the entry-exit site of proximal 601-nucleosomes of the di-nucleosome. The authors did not specify the linker sequences used in this study, although they changed the nucleosomal DNA sequence from 601 to 603 DNA. A linker with palindromic sequence can be used to test the effect of DNA sequence on the preferential digestion of Tn5.
2. The authors used di-nucleosomes without outer linker in this study. However, as the entry and exit site of nucleosomal DNA are asymmetric on the di-nucleosome, especially when considering steric hindrance of nucleosomes, they would show different effects on the integration of Tn5. This is especially interesting when considering one nucleosome adjacent to a nucleosome-free region *in vivo*. Thus, the authors should analyze the integration pattern of Tn5 on di-nucleosomes with different lengths of outer linker DNA. Alternatively, the authors can perform Tn5 integration experiments on mononucleosomal template with different lengths of linker DNAs.
3. At Page 13 of 26, the method "4.5. Sequence analysis", the authors stated that "The resulting DNA fragments were purified to exclude fragments shorter than 150 base-pairs" after the second adaptor ligation. Thus, it is possible that fragments with cutting sites inside the nucleosome be left out. Alicia N. Schep et al. (Genome Research, 2015) has reported that Tn5 could cut into the nucleosome. Thus, it is important to retain this information to compare the cutting efficiency inside and outside of nucleosome. Moreover, Alicia N. Schep et al observed periodicity of Tn5 integration at the boundary of the nucleosome, which is different from the reported specific integration at the entry-exit site the nucleosome. It's possible that the integration pattern of Tn5 on the nucleosome would depend on the extent of digestion. Thus, it would be important to sequence fragments from different digestion extend. Besides, the authors did not point out the digestion condition they performed sequencing on.

Minor concerns:

1. In Page 9 of 26, the authors stated "Nucleosome formation induces DNA negative supercoiling, and may assist the Tn5 transposase integration at specific nucleosome sites, as observed in the present study". It is possibly true that the negative supercoiling induced by nucleosome assembly may affect Tn5 transposase integration *in vivo*. However, it is hardly the case observed in this study, since the di-nucleosomes used in this study are assembled with linear DNA as described in methods "4.1. Preparation of dinucleosomes". To test the effect of DNA supercoiling on the integration of Tn5, circular plasmid DNA could be used for nucleosome assembly.

Decision letter (RSOB-19-0116.R0)

04-Jul-2019

Dear Professor Kurumizaka

We are pleased to inform you that your manuscript RSOB-19-0116 entitled "Biochemical analysis of chromatin targeting by Tn5 transposase" has been accepted by the Editor for publication in Open Biology. The reviewer(s) have recommended publication, but also suggest some minor revisions to your manuscript. Therefore, we invite you to respond to the reviewer(s)' comments and revise your manuscript.

Please submit the revised version of your manuscript within 14 days. If you do not think you will be able to meet this date please let us know immediately and we can extend this deadline for you.

- 1) A text file of the manuscript (doc, txt, rtf or tex), including the references, tables (including captions) and figure captions. Please remove any tracked changes from the text before submission. PDF files are not an accepted format for the "Main Document".
- 2) A separate electronic file of each figure (tiff, EPS or print-quality PDF preferred). The format should be produced directly from original creation package, or original software format. Please note that PowerPoint files are not accepted.
- 3) Electronic supplementary material: this should be contained in a separate file from the main text and meet our ESM criteria (see <https://royalsocietypublishing.org/rsob/for-authors#question3>). All supplementary materials accompanying an accepted article will be treated as in their final form. They will be published alongside the paper on the journal website and posted on the online figshare repository. Files on figshare will be made available approximately one week before the accompanying article so that the supplementary material can be attributed a unique DOI.

Online supplementary material will also carry the title and description provided during submission, so please ensure these are accurate and informative. Note that the Royal Society will not edit or typeset supplementary material and it will be hosted as provided. Please ensure that the supplementary material includes the paper details (authors, title, journal name, article DOI). Your article DOI will be 10.1098/rsob.2016[last 4 digits of e.g. 10.1098/rsob.20160049].

- 4) A media summary: a short non-technical summary (up to 100 words) of the key findings/importance of your manuscript. Please try to write in simple English, avoid jargon, explain the importance of the topic, outline the main implications and describe why this topic is newsworthy.

Images

Data-Sharing

It is a condition of publication that data supporting your paper are made available. Data should be made available either in the electronic supplementary material or through an appropriate repository. Details of how to access data should be included in your paper. Please see <http://royalsocietypublishing.org/site/authors/policy.xhtml#question6> for more details.

Data accessibility section

Sincerely,

The Open Biology Team

<mailto:openbiology@royalsociety.org>

Reviewer(s)' Comments to Author:

Referee: 1

Comments to the Author(s)

This manuscript presents an investigation into Tn5 transposase targetting on defined dinucleosomal templates. It links the high quality nucleosome preparation skills of the Kurumizaka laboratory to create a substrate for analysis of in vitro Tn5 attack on nucleosomes with a next generation sequencing readout.

The methods and results of this study are clear and well-described. The title could be slightly edited to "Biochemical analysis of nucleosome targeting by Tn5 transposase" to better reflect the nature of the work.

However, the depth of interpretation provided is very weak and should make more effort to address the question of dependence of Tn5 site specificity on DNA sequence versus nucleosome structure. This is very important because the extreme bias the authors report of incorporation at only 1-3 sites impacts strongly on broader use and interpretation of ATAC-seq experiments.

Firstly, the relative enhancement/protection by the histone octamer is highly relevant to the interpretation. DNA sequence preferences of Tn5 on free DNA have been published and the authors probably have NGS data for the free DNA substrates they show. Inspection of free DNA bands in figures 1C and 3B suggests that the sites identified as targets with the nucleosome substrate may already be strongly preferred in free DNA. A more detailed analysis could also incorporate data from *S. cerevisiae* mapping by the Greenleaf group.

A related aspect is the spread of incorporation. Figure 1E and the text seems to imply that Tn5 insertion is a single base sites, whereas figure 1D appears shows spread over at least 3 bases on

601-15, and figure 4 shows even more spread on 603. A more detailed description of this is needed.

Secondly, more detailed structural consideration should be included. Since the structure of the 601 containing nucleosome and Tn5 transposase are known, it is reasonable to provide a simple docking schematic to assess the geometry and spatial features observed. The relevance of the co-crystal structure of a nucleosome with prototype foamy virus integrase should be discussed, but has not even been cited. Such a discussion does not need to overshadow the impact of a nucleosome-Tn5 structure that the authors may be attempting to determine.

One smaller issue is over-referencing in blocks of citations: Of the 65 total references, 58 (89%) are cited in the Introduction. On page 2 line 26 there is a block of 13 references for genome-wide nucleosome mapping, on page 3 line 1 there are 4 references for generic genome functionality, on page 3 line 9 there are 5 references for DNaseI mapping, and on page 4 line 15 there are 5 + 12 references for applications of ATAC-seq. In total, these 4 blocks have 39 citations accounting for 60% of all references. With due respect to the cited primary research, many of these references could be subsumed by citing a summary review and a few key primary papers.

There do not seem to be any figure legends for the supplementary figures.

Referee: 2

Comments to the Author(s)

Tn5 transposase has been widely used for the mapping of accessible chromatin regions, which are usually the transcription binding sites. However, little attention has been paid on the effect of nucleosome/chromatin structure on the integration of Tn5. In the manuscript titled as "Biochemical analysis of chromatin targeting by Tn5 transposase", the authors explicitly analyzed the effects of in vitro assembled di-nucleosomes with different linker length on the integration of Tn5. They found that Tn5 transposase minimally cleaved the di-nucleosome without a linker DNA and Tn5 transposase preferentially targets near the entry-exit sites of the proximal nucleosomal DNA. In addition to the entry-exit sites of the nucleosome, Tn5 transposase would additionally target the middle of the linker DNA when the linker length of di-nucleosome extended to 30 bp. In conclusion, this study provides useful insights into the mechanisms by which nucleosomes affect Tn5 integration. However, several concerns need to be addressed before this manuscript could be accepted for publication.

Major concerns:

1. The authors observed preferential integration of Tn5 at the entry-exit site of proximal 601-nucleosomes of the di-nucleosome. The authors did not specify the linker sequences used in this study, although they changed the nucleosomal DNA sequence from 601 to 603 DNA. A linker with palindromic sequence can be used to test the effect of DNA sequence on the preferential digestion of Tn5.
2. The authors used di-nucleosomes without outer linker in this study. However, as the entry and exit site of nucleosomal DNA are asymmetric on the di-nucleosome, especially when considering steric hindrance of nucleosomes, they would show different effects on the integration of Tn5. This is especially interesting when considering one nucleosome adjacent to a nucleosome-free region in vivo. Thus, the authors should analysis the integration pattern of Tn5 on di-nucleosomes with different length of outer linker DNA. Alternatively, the authors can performed Tn5 integration experiments on mononucleosomal template with different length of linker DNAs.
3. At Page 13 of 26, the method "4.5. Sequence analysis", the authors stated that "The resulting DNA fragments were purified to exclude fragments shorter than 150 base-pairs" after the second

adaptor ligation. Thus, it is possibly that fragments with cutting sites inside the nucleosome be left out. Alicia N. Schep et al. (Genome Research, 2015) has reported that Tn5 could cut into the nucleosome. Thus, it is important to retain this information to compare the cutting efficiency inside and outside of nucleosome. Moreover, Alicia N. Schep et al observed periodicity of Tn5 integration at the boundary of the nucleosome, which is different from the reported specific integration at the entry-exit site the nucleosome. It's possibly that the integration pattern of Tn5 on the nucleosome would depend on the extend of digestion. Thus, it would be important to sequence fragments from different digestion extend. Besides, the authors did not point out the digestion condition they performed sequencing on.

Minor concerns:

1. In Page 9 of 26, the authors stated "Nucleosome formation induces DNA negative supercoiling, and may assist the Tn5 transposase integration at specific nucleosome sites, as observed in the present study". It is possibly true that the negative supercoiling induced by nucleosome assembly may affect Tn5 transposase integration in vivo. However, it is hardly the case observed in this study, since the di-nucleosomes used in this study are assembled with linear DNA as described in methods "4.1. Preparation of dinucleosomes". To test the effect of DNA supercoiling on the integration of Tn5, circular plasmid DNA could be used for nucleosome assembly.

Author's Response to Decision Letter for (RSOB-19-0116.R0)

See Appendix A.

Decision letter (RSOB-19-0116.R1)

22-Jul-2019

Dear Professor Kurumizaka

We are pleased to inform you that your manuscript entitled "Biochemical analysis of nucleosome targeting by Tn5 transposase" has been accepted by the Editor for publication in Open Biology.

Sincerely,

The Open Biology Team
mailto: openbiology@royalsociety.org

Appendix A

Reviewers' comments:

Referee: 1

Comment)

The methods and results of this study are clear and well-described. The title could be slightly edited to "Biochemical analysis of nucleosome targeting by Tn5 transposase" to better reflect the nature of the work.

Reply)

Thank you for this suggestion. In the revised manuscript, we changed the title to "Biochemical analysis of nucleosome targeting by Tn5 transposase".

Comment)

*Firstly, the relative enhancement/protection by the histone octamer is highly relevant to the interpretation. DNA sequence preferences of Tn5 on free DNA have been published and the authors probably have NGS data for the free DNA substrates they show. Inspection of free DNA bands in figures 1C and 3B suggests that the sites identified as targets with the nucleosome substrate may already be strongly preferred in free DNA. A more detailed analysis could also incorporate data from *S. cerevisiae* mapping by the Greenleaf group.*

Reply)

Thank you very much for this insightful comment. According to this reviewer's comment, we have added the NGS data for the free DNA substrates in the new Supplementary Table S1. In the presence of the free DNA substrates, many target sites of Tn5 could not be annotated on a single location, because of the tandemly repetitive sequences for the dinucleosome template DNAs. In contrast, nucleosome formation masked the target sites in a nucleosome. This technical problem makes it difficult to provide an accurate graphical representation for the target sites in free DNAs. Therefore, we have listed the top five cleavage sites (highest counts in the NGS analysis) for the free DNAs in Supplementary Table S1. In the presence of the 601 dinucleosome substrate, the 5'-140 base position (upper strand) was one of the major target sites for Tn5 transposase. This site was also observed as one of the major target sites for Tn5 transposase in the free DNA substrate, as listed in Supplementary Table S1, suggesting

that the 140 base position appears to be a preferred sequence for Tn5 transposase. This fact does not affect our conclusion, because this is not the case in the experiments with the 603 DNA sequence. In the revised manuscript, we added a description of these facts in the results section (p.7, ll.2-5), and intensively discussed this point in the discussion section (p.8, l.24-p.9, l.3).

Comment)

A related aspect is the spread of incorporation. Figure 1E and the text seems to imply that Tn5 insertion is a single base sites, whereas figure 1D appears shows spread over at least 3 bases on 601-15, and figure 4 shows even more spread on 603. A more detailed description of this is needed.

Reply)

In the revised manuscript, we listed the detailed cleavage sites of the 601-15, 601-30, 603-15, and 603-30 dinucleosome substrates in Supplementary Table S1.

Comment)

Secondly, more detailed structural consideration should be included. Since the structure of the 601 containing nucleosome and Tn5 transposase are known, it is reasonable to provide a simple docking schematic to assess the geometry and spatial features observed. The relevance of the co-crystal structure of a nucleosome with prototype foamy virus integrase should be discussed, but has not even been cited. Such a discussion does not need to overshadow the impact of a nucleosome-Tn5 structure that the authors may be attempting to determine.

Reply)

As this reviewer suggested, in the revised manuscript, we have cited the published report for a prototype foamy virus integrase complexed with a nucleosome (Maskell et al., Nature 2015), and discussed the integration mechanism of the presumed binding sites of Tn5 and the foamy virus integrase in the revised manuscript (p.10, ll.14-22).

Comment)

One smaller issue is over-referencing in blocks of citations: Of the 65 total references,

58 (89%) are cited in the Introduction. On page 2 line 26 there is a block of 13 references for genome-wide nucleosome mapping, on page 3 line 1 there are 4 references for generic genome functionality, on page 3 line 9 there are 5 references for DNaseI mapping, and on page 4 line 15 there are 5 + 12 references for applications of ATAC-seq. In total, these 4 blocks have 39 citations accounting for 60% of all references. With due respect to the cited primary research, many of these references could be subsumed by citing a summary review and a few key primary papers.

Reply)

According to this reviewer's comment, we re-selected the references in the revised manuscript.

Comment)

There do not seem to be any figure legends for the supplementary figures.

Reply)

Thank you for this comment. We have added the figure legends in the supplementary figure files.

Referee: 2

Major concerns:

Comment)

The authors observed preferential integration of Tn5 at the entry-exit site of proximal 601-nucleosomes of the di-nucleosome. The authors did not specify the linker sequences used in this study, although they changed the nucleosomal DNA sequence from 601 to 603 DNA. A linker with palindromic sequence can be used to test the effect of DNA sequence on the preferential digestion of Tn5.

Reply)

We thank the reviewer for this comment. We have presented the DNA sequences for the dinucleosome substrates used in this study in Supplementary Fig. S4.

Comment)

The authors used di-nucleosomes without outer linker in this study. However, as the entry and exit site of nucleosomal DNA are asymmetric on the di-nucleosome, especially when considering steric hindrance of nucleosomes, they would show different effects on the integration of Tn5. This is especially interesting when considering one nucleosome adjacent to a nucleosome-free region in vivo. Thus, the authors should analysis the integration pattern of Tn5 on di-nucleosomes with different length of outer linker DNA. Alternatively, the authors can performed Tn5 integration experiments on mononucleosomal template with different length of linker DNAs.

Reply)

We appreciate this referee's comment. In this study, we focused on the effect of the linker DNA between two nucleosomes on the integration reaction. Therefore, the question of how the outer linker DNAs affect the integration reaction by Tn5 transposase in chromatin is beyond the scope of the present research. We will study this referee's point, as an interesting future topic.

Comment)

At Page 13 of 26, the method "4.5. Sequence analysis", the authors stated that "The resulting DNA fragments were purified to exclude fragments shorter than 150 base-pairs" after the second adaptor ligation. Thus, it is possibly that fragments with cutting sites inside the nucleosome be left out. Alicia N. Schep et al. (Genome Research, 2015) has reported that Tn5 could cut into the nucleosome. Thus, it is important to retain this information to compare the cutting efficiency inside and outside of nucleosome.

Reply)

As shown in the gel-based analysis of the dinucleosome substrates with a 15 base-pair linker DNA, the transposase reaction by the Tn5-DNA complex results in two distinct DNA fragments, indicating that Tn5 transposase cleaved the dinucleosomes at a single site (Figures 1C, 2B, and 3B). Therefore, the cleavage site can be mapped by a sequence analysis with only the longer DNA. Our mapping data are consistent with the previous genomics study reported by Schep et al. (Genome Research, 2015).

Comment)

Moreover, Alicia N. Schep et al observed periodicity of Tn5 integration at the boundary of the nucleosome, which is different from the reported specific integration at the entry-exit site the nucleosome. It's possibly that the integration pattern of Tn5 on the nucleosome would depend on the extend of digestion. Thus, it would be important to sequence fragments from different digestion extend. Besides, the authors did not point out the digestion condition they performed sequencing on.

Reply)

When the dinucleosome was treated with the Tn5-DNA complex for longer times (for example, 45 min and 90 min, Figure 1C lanes 10-12), we observed the weak but clear periodic digestion pattern on the non-denaturing polyacrylamide gel. These fragments may be products corresponding to the periodic digests by Tn5 transposase, as previously reported by Schep et al. This point is described in the revised manuscript (p.5, ll.14-17).

Minor concerns:

Comment)

In Page 9 of 26, the authors stated “Nucleosome formation induces DNA negative supercoiling, and may assist the Tn5 transposase integration at specific nucleosome sites, as observed in the present study”. It is possibly true that the negative supercoiling induced by nucleosome assembly may affect Tn5 transposase integration in vivo. However, it is hardly the case observed in this study, since the di-nucleosomes used in this study are assembled with linear DNA as described in methods “4.1. Preparation of dinucleosomes”. To test the effect of DNA supercoiling on the integration of Tn5, circular plasmid DNA could be used for nucleosome assembly.

Reply)

Thank you very much. According to this reviewer's suggestion, in the revised manuscript, we removed the related paragraph discussing the DNA supercoiling in the discussion section.